# Rumen Fermentation—Microbiota—Host Gene Expression Interactions to Reveal the Adaptability of Tibetan Sheep in Different Periods

**DOI:** 10.3390/ani11123529

**Published:** 2021-12-10

**Authors:** Weibing Lv, Xiu Liu, Yuzhu Sha, Hao Shi, Hong Wei, Yuzhu Luo, Jiqing Wang, Shaobin Li, Jiang Hu, Xinyu Guo, Xiaoning Pu

**Affiliations:** College of Animal Science and Technology/Gansu Key Laboratory of Herbivorous Animal Biotechnology, Gansu Agricultural University, Lanzhou 730070, China; LWB18794850948@163.com (W.L.); shayz@st.gsau.edu.cn (Y.S.); shih@st.gsau.edu.cn (H.S.); weih@st.gsau.edu.cn (H.W.); luoyz@gsau.edu.cn (Y.L.); wangjq@gsau.edu.cn (J.W.); lisb@gsau.edu.cn (S.L.); huj@gsau.edu.cn (J.H.); guoxinyu6662021@163.com (X.G.); pxn18893478406@163.com (X.P.)

**Keywords:** rumen fermentation, microbiota, 16S rRNA, gene expression, Tibetan sheep

## Abstract

**Simple Summary:**

The Qinghai-Tibet Plateau has a unique ecological environment, involving high altitude, low oxygen levels, strong ultraviolet rays, and severe imbalances in seasonal forage supply, which poses a serious threat to the livestock that feeds on natural pastures to maintain their survival. We have carried out a long-term follow-up study on rumen fermentation characteristics, the microbiota, and rumen epithelial gene expression of local Tibetan sheep. Correlation analysis showed that there were interactions among rumen fermentation characteristics, the microbiota, and host gene expression, mainly by adjusting the amino acid metabolism pathway and energy metabolism pathway to improve energy utilization. At the same time, we adjusted the balance of the rumen “core microbiota”, which was regulated to promote the development of rumen and maintain the homeostasis of rumen environment (which relies Tibetan sheep can better adapt to the harsh environment in different periods of the Qinghai-Tibet Plateau). This provides a theoretical basis for the breeding and management of Tibetan sheep on the Qinghai-Tibet Plateau.

**Abstract:**

As an important ruminant on the Qinghai-Tibet Plateau, Tibetan sheep can maintain their population reproduction rate in the harsh high-altitude environment of low temperature and low oxygen, which relies on their special plateau adaptations mechanism that they have formed for a long time. Microbiomes (known as “second genomes”) are closely related to the nutrient absorption, adaptability, and health of the host. In this study, rumen fermentation characteristics, the microbiota, and rumen epithelial gene expression of Tibetan sheep in various months were analyzed. The results show that the rumen fermentation characteristics of Tibetan sheep differed in different months. The total SCFAs (short-chain fatty acids), acetate, propionate, and butyrate concentrations were highest in October and lowest in June. The CL (cellulase) activity was highest in February, while the ACX (acid xylanase) activity was highest in April. In addition, the diversity and abundance of rumen microbes differed in different months. Bacteroidetes (53.4%) and Firmicutes (27.4%) were the dominant phyla. *Prevotella_1* and *Rikenellaceae_RC9_gut_group* were the dominant genera. The abundance of *Prevotella_1* was highest in June (27.8%) and lowest in December (17.8%). In addition, the expression of *CLAUDIN4* (Claudin-4) and *ZO1* (Zonula occludens 1) was significantly higher in April than in August and December, while the expression of *SGLT1* (Sodium glucose linked transporter 1) was highest in August. Correlation analysis showed that there were interactions among rumen fermentation characteristics, the microbiota, and host gene expression, mainly by adjusting the amino acid metabolism pathway and energy metabolism pathway to improve energy utilization. At the same time, we adjusted the balance of the rumen “core microbiota” to promote the development of rumen and maintain the homeostasis of rumen environment, which makes Tibetan sheep better able to adapt to the harsh environment in different periods of the Qinghai-Tibet Plateau.

## 1. Introduction

The Qinghai-Tibet Plateau is the world’s highest and largest plateau and the only plateau used for grazing in four seasons. It has a unique ecological environment, involving high altitude, low oxygen levels, strong ultraviolet rays, and severe imbalances in seasonal forage supply [1]. Livestock living on this plateau have special high-altitude adaptations that allow them to maintain their normal reproduction rate. The green-up stage (from early April to May), green grass stage (from June to September) and withered grass stage (from September to March of the following year) of plateau grasslands reflect the changes in ambient temperature. The phenological changes in the plants of plateau grasslands significantly influence the grassland ecosystem, grazing livestock, and their gut microbes [2]. The nutrient composition of the forage grass in the Qinghai-Tibet Plateau pastures changes with the season. Locally grazing Tibetan sheep mainly obtain nutrients by eating natural forages. During the withered grass stage, Tibetan sheep obtain nutrients only from withered grass, which allows them to maintain their basic metabolic and herd reproduction. They provide local herders with meat, wool, fur, fuel (animal dung as fuel), and other daily necessities, making them indispensable economic animals for local herders.

The types and abundances of microbes in the host are associated with the host growth performance, physiological phenotype, and health status [3]. The gut microbiota regulates immune processes and metabolism using microbial metabolites (such as short-chain fatty acids [SCFAs]) to maintain homeostasis, regulate immune processes, and protect host health [4]. Changes in the gut microbiota composition or function may induce diseases in the host [5]. The rumen microbiota in ruminants is responsible for decomposing plant-derived forages to produce nutrients that can be directly used by the host. The microbes participate in host metabolism and exchange nutrients and information with the host. The microbiota composition is affected by various factors such as diet, species, host physiology, and the environment, among which diet is the most important factor [6]. The microbiota composition determines the specific fermentation pathways in the rumen [7]. The intestinal microbes have strong symbiotic relationships with protozoa and fungi [8], and they co-evolved with the host, playing important roles in forage fermentation and host energy supply [8]. The gut microbiota can help the host to decompose cellulose, hemicellulose, and other substances that the host cannot directly use, thereby providing the host with energy and nutrients [9].

The rumen epithelium is a unique place where the host interacts with its microbiota, and the microbiota affects the host’s net nutrient utilization by providing nutrients [10]. Intestinal epithelial cells (IECs) are considered to be immune cells [11], and microbiota metabolites are closely related to these epithelial cells. For example, butyrate promotes intestinal homeostasis by regulating IECs [12]. The rumen epithelial tissue of ruminants has complex tight junctions (TJs) composed of intercellular protein complexes forming selective barriers that interact with the rumen microbiota [13]. *CLAUDIN4* (Claudin-4) is a key TJ protein, playing a key role in the barrier function [14]. *ZO1* (Zonula occludens 1) was the first TJ protein to be discovered, and it colocalizes with *CLAUDIN4* in the apical intercellular space between epithelial cells, playing a role in TJ formation and regulation [15].

Rumen microbes can metabolize complex carbohydrates into d-glucose, which is broken down into SCFAs. High-affinity *SGLT1* (Sodium glucose linked transporter 1) in the apical side of the sheep rumen epithelium can actively transport d-glucose from the rumen to the blood, so that the rumen epithelium maintains the ability to absorb glucose [16]. There is a close relationship between the rumen microbiota and the expression of host nutrient absorption- and barrier-related genes, and the rumen microbiota influences the host’s nutrient absorption, energy utilization, and ruminal homeostasis.

Our team’s previous research found that there are significant differences between the rumen microbiota of grazing Tibetan sheep in cold and warm seasons and the expression of genes related to nutrient absorption and rumen epithelial barrier, and rumen microbes are significantly related to some fermentation products and rumen epithelial genes [17]. Therefore, this study comprehensively analyzed the effects of the interactions among the rumen fermentation, the microbiota, and host gene expression on the adaptation of Tibetan sheep to the plateau, providing a theoretical basis regarding the adaptation mechanisms of Tibetan sheep on the plateau.

## 2. Materials and Methods

### 2.1. Experimental Design and Sample Collection

In January 2019, we selected 15 healthy Tibetan sheep ewes (one year old, weight = 35.12 ± 1.43 kg) obtained from a farm in Gannan Tibetan Autonomous Prefecture (Gansu Province, China), which used local traditional natural grazing management, did not undergo any supplementary feeding and was located at an altitude of 3300 m. In different months (in February, April, June, August, October, and December), the rumen fluid (50 mL) was collected with a sheep gastric tube rumen sampler in the morning and filtered with 4 layers of sterile gauze, and were immediately frozen in liquid nitrogen used for 16S rRNA sequencing and determination of SCFAs, NH_3_-N content, CL, and ACX activities (stored at −80 °C).

The slaughter trials were carried out in April (returning green stage), August (green grass stage), and December (withered grass stage), and three Tibetan sheep were randomly slaughtered each time. Dissected immediately after slaughter (the jugular vein bloodletting), and rumen epithelial tissue samples were acquired by dissecting a small piece of rumen abdominal sac, quickly removing rumen contents by rinsing with phosphate-buffered saline (PBS), then separating the epithelial tissue with blunt scissors. Tissue samples were placed in liquid nitrogen and stored at −80 °C refrigerator for subsequent RNA extraction.

### 2.2. Determination of Rumen Fermentation Parameters

SCFAs were determined with a gas chromatograph (GC-2010 plus; Shimadzu, Japan). The internal standard method was adopted, using 2-ethyl butyric acid (2EB) as the internal standard. The chromatographic column was an AT-FFAP (50 m × 0.32 mm × 0.25 μm) capillary column. The column temperature was maintained at 60 °C for 1 min, then increased to 115 °C at 5 °C/min without reservation, and then increased to 180 °C at 15 °C/min. The detector temperature was 260 °C and the injector temperature was 250 °C. Measure the content of NH_3_-N in rumen fluid by spectrophotometer colorimetry. The CL (cellulase) and ACX (acid xylanase) activities in the rumen fluid were determined with a CL activity assay kit and an ACX assay kit (Suzhou Keming Biotechnology Co., Ltd., Shanghai, China), respectively.

### 2.3. Microbial DNA Extraction and 16S rRNA Sequencing

Total DNA was isolated from each rumen sample using MN NucleoSpin 96 Soil kit (Macherey-Nagel, Düren, Germany). Use universal primers (338F: 5′-ACTCCTACGGGAGGCAGCAG-3′ and 806R: 5′-GGACTACHVGGGTWTCTAAT-3′) to PCR amplified of the V3-V4 region of the 16S rRNA gene hypervariable region [18], The two-step library construction method was used to construct and sequence the rumen microbes [19], All amplified products were sequenced and analyzed on an Illumina MiSeq platform (Illumina, San Diego, CA, USA).

### 2.4. Analysis of mRNA Levels in Rumen Epithelial Tissue

Trizol reagent method was used to extract RNA from rumen epithelial tissue of Tibetan sheep (TransGen, Shenyang, China). Use an ultra-micro spectrophotometer (Therm Nano Drop-2000) to determine the concentration and purity of RNA. cDNA synthesis was carried out by reverse transcription kit (HiScript^®^ II Q RT SuperMix for qPCR; Nanjing, China). Use Primer 5.0 software to design primers for *SGLT1*, *CLAUDIN4*, and *ZO1* genes. The internal reference gene is *β-actin*. See Table 1 for primer information. Use Applied Biosystems Q6 real-time fluorescent quantitative PCR instrument for relative quantification of related genes, using a 20 µL reaction system, reaction conditions: 95 °C pre-denaturation for 30 s; cyclic reaction at 95 °C for 10 s, 60 °C for 30 s, 40 cycles; dissolution curve conditions (95 °C for 15 s, 60 °C for 60 s, 95 °C for 15 s). The *β-actin* was used as the internal reference gene for calibration, and the data analysis adopted the 2^−∆∆CT^ method [20].

### 2.5. Bioinformatics Analysis

The quality of the original sequencing data was evaluated, and the original sequencing reads were denoised and subjected to paired-end read merging (FLASH, version 1.2.11), quality filtering (Trimmomatic, version 0.33), and chimera removal (UCHIME, version 8.1). Usearch software (version 10.0) was employed to cluster the high-quality effective tags, based on 97% similarity, using the 2013 Greengenes Ribosome Database (version 13.8). The OTUs were filtered [21], using 0.005% of all sequences sequenced as the threshold. Using the SILVA database (bacterial 16S), species annotation and taxonomic analysis of OTU were conducted.

Using Mothur (version 1.30), the OTUs were subjected to an alpha diversity analysis (including Shannon index). Rarefaction and species relative abundance accumulation curves were also constructed. Beta diversity analysis was performed using QIIME, with PCoA, non-metric multi-dimensional scaling (NMDS), and ANOSIM being used to compare the species diversity among samples collected in different months. LEfSe analysis was used to identify the biomarkers related to the samples obtained in each month.

Finally, PICRUSt software was used to compare the functional differences related to microbiota species composition (based on 16S sequencing data) among samples obtained in different months. First, a differential KEGG pathway analysis was used to identify the differences in the metabolic pathways related to the functional genes of the microbiota in different months. Second, a differential COG function prediction analysis was used to predict the differences in the protein function of the prokaryotes in different months.

### 2.6. Statistical Analysis

Aanlysis of Covariance (ANCOVA) in SPSS software (version 24.0, SPSS Inc., Chicago, IL, USA) was used to analyze differences in rumen fermentation parameters (SCFAs, NH_3_-N content, CL and ACX activities), alpha diversity index (Ace Index, Chao1 Index, Shannon Index, and Simpson Index) and related gene expression (*SGLT1*, *CLAUDIN4,* and *ZO1*) in the rumen of Tibetan sheep at different stages. The normality of the data was tested by Shapiro-Wilk test and the homogeneity of variance was tested by Levene test. LEfSe method was used to evaluate the differences of microbial communities, and the LDA score > 4. The analysis result data are all expressed as mean ± standard deviation (M ± SD), and *p* < 0.05 indicated a significant difference. The Spearman correlation test was used to analyze the correlation between the bacterial genera (top 20 species with high abundances of their corresponding genera) and fermentation parameters (SCFAs and NH_3_-N levels, and CL and ACX activities) and gene expression genes.

## 3. Results

### 3.1. Rumen Fermentation Parameters

The rumen fermentation parameters in the Tibetan sheep are shown in Table 2. The SCFAs concentrations in the rumen of Tibetan sheep differed in different months. The total SCFAs concentration was significantly higher in October than in other months (*p* < 0.05). The acetate, propionate, and butyrate concentrations peaked in different months, and the sum of the three accounted for >83.1% of the total SCFAs. The acetate concentration was highest in October and lowest in June, and significantly higher in October and December than in other months (*p* < 0.05) and significantly lower in June than in other months (*p* < 0.05). The propionate concentration was significantly higher in October than in other months (*p* < 0.05), and lowest in June. The butyrate concentration was significantly higher in August and October than in other months (*p* < 0.05), and significantly lower in June than in other months (*p* < 0.05). The isobutyrate, isovalerate, and valerate concentrations were significantly higher in February and April than in other months (*p* < 0.05), and lowest in December. The acetate:propionate (A:P) was significantly higher in December than in other months (*p* < 0.05), and lowest in February. In addition, the NH_3_-N level in the rumen fluid was significantly higher in June and August than in other months, and was lowest in February. The CL and ACX activities in the rumen fluid also differed in different months. More specifically, CL activity was highest in February (when it was significantly higher than in August and October), and ACX activity was highest in April (and it was significantly higher in both April and June than in other months) and lowest in August.

### 3.2. Characteristics of Rumen Microflora in Different Months

#### 3.2.1. Rumen Microbial 16S rRNA Sequencing Results and Alpha and Beta Diversity

16S rRNA sequencing was performed on 36 samples obtained in different months, and 2,879,764 pairs of reads were obtained. After paired-end read merging and filtering, 2,773,869 clean tags were produced. Each sample produced at least 73,683 clean tags, with a mean of 77,052, and a mean sequence length of 419 bp. Using Usearch software to the cluster tags (based on a 97% similarity level), 1223 OTUs were obtained, comprising 1150, 1168, 1064, 1149, 1155, and 1168 in February, April, June, August, October, and December, respectively. There were 941 OTUs that were shared by samples from Tibetan sheep rumens obtained in different months, and one unique OTU was found in the December samples, as shown in the Venn diagram in Figure 1A. The rarefaction curve (Figure 1B) plateaued at 20,000 reads, that is, the number of species did not significantly increase with increases in the number of tags sampled, indicating saturation. The sequencing coverage rate reached > 99.6%, with the sequencing data authentically and reliably reflecting the samples (Table 3).

The number of OTUs in each month was highest in October and lowest in June, and significantly higher in August and October than in other months (*p* < 0.05), and significantly lower in June than in other months (*p* < 0.05). The Shannon index was significantly higher in August, October, and April than in other months (*p* < 0.05). The Simpson index exhibited no significant differences in different months. The abundance-based coverage estimator (ACE) and Chao1 index (representing the abundances of rumen microbes) was significantly higher in August and October than in other months (*p* < 0.05), and lowest in June (Table 3). Principal coordinates analysis (PCoA) indicated obvious clustering related to the month of sample collection, with the August and October samples having similar rumen microbial diversity, the February and April samples having similar rumen microbial diversity, and the June samples having quite different rumen microbial diversity compared to the other months (Figure 1C). Analysis of similarities (ANOSIM) indicated that the difference regarding samples from different months was significantly greater than the difference regarding samples from the same month, the reliability of the sequencing data was high, and subsequent analysis could be performed (Figure 1D).

#### 3.2.2. Numbers of Phyla and Genera and Differential Taxa

There were 17 phyla, 26 classes, 38 orders, 59 families, 149 genera, and 163 species. Bacteroidetes (>53.4%) and Firmicutes (>27.4%) were the dominant bacterial phyla, accounting for >83.3% of the total abundance in each month (Figure 2A). The Firmicutes:Bacteroidetes (F:B) ratio was significantly higher in August (0.619) than in April (0.500) and December (0.483). The abundance of Patescibacteria was >1% in each month, and there were significant differences in different months (*p* < 0.05), with the highest abundance in February (8.17%) and the lowest in June (1.09%). Furthermore, as shown in the rank sum test chart, there were 12 different species at the phylum level (Figure 2B). In particular, Firmicutes was significantly higher in June than in February and December (*p* < 0.05). Patescibacteria was significantly lower in June than in February and December (*p* < 0.05), while Kiritimatiellaeota exhibited the opposite trend. Elusimicrobia and Cyanobacteria were both significantly lower in June than in February and April (*p* < 0.05). Bacteroidetes had the highest abundance but exhibited no significant differences in different months.

There were 108 different species in 149 genera (*p* < 0.05). The abundances of *Prevotella_1*, *Rikenellaceae_RC9_gut_group*, *Christensenellaceae_R-7_group*, *Ruminococcaceae_NK4A214_group*, and *Prevotellaceae_UCG-001* were > 1% in each month (Figure 2C). The dominant bacteria genera in each month were Prevotella_1 (>17.8%; highest in June [27.8%] and lowest in December [17.8%]) and Rikenellaceae_RC9_gut_group (>7.2%; highest in December [25.8%] and lowest in August [7.2%]).

There were 12, 5, and 12 significantly differential species between April and August (rejuvenating grass vs. green grass), April and December (rejuvenating grass vs. withered grass), and August and December (green grass vs. withered grass), respectively, were found. According to the rank sum test chart (showing the top 20 species with the lowest *p* values at the genus level), *Ruminiclostridium_9*, *Family_XIII_UCG-001*, and *Acinetobacter* were significantly different between April and August; there were no significantly different genera between April and December; and *Rikenellaceae_RC9_gut_group*, *DNF00809*, *Coprococcus_1*, and *Acinetobacter* were significantly different between August and December (Figure 2D). In addition, *Butyrivibrio_2*, *Moryella* and *Shuttleworthia* were significantly higher in June than in February and December (*p* < 0.05); *Olsenella* was significantly lower in February than in August and October; *Anaeroplasma* and *Sphaerochaeta* were significantly lower in June than in February and April; and *Acetitomaculum* was significantly higher in October than in February and December. Random forest analysis at the genus level showed that *Lachnospiraceae_NK3A20_group* was the most important for sample classification, followed by *DNF00809*, *Rikenellaceae_RC9_gut_group*, and *FD2005* (Figure 3).

#### 3.2.3. Microbe Biomarkers in Different Months

To identify biomarkers that could significantly differentiate between different months, Linear discriminant analysis (LDA) effect size (LEfSe) analysis was performed (Figure 4). Based on Linear discriminant analysis (LDA) scores > 4, the identified biomarkers for February, April, June, August, October, and December were Patescibacteria; *uncultured_bacterium_f_F082*; Firmicutes, Synergistetes, *Prevotella_1*, *Butyrivibrio_2*, *Succiniclasticum*, *Ruminococcaceae_NK4A214_group*, and *Fretibacterium*; *Prevotellaceae_UCG-003*; *Prevotellaceae_UCG-001* and *Prevotella_9*; and Bacteroidetes, *Christensenellaceae_R-7_group*, and *Rikenellaceae_RC9_gut_group*, respectively.

### 3.3. Prediction of Rumen Microbial Gene Function

Next, PICRUSt software was used for gene function prediction based on the ruminal microbe 16S rRNA sequencing data.

Among the 46 Kyoto Encyclopedia of Genes and Genomes (KEGG) gene families and 25 Clusters of Orthologous Groups of proteins (COG) functional genes identified in the April and August samples (rejuvenating and green grass stages, respectively), METABOLISM pathway-related genes accounted for >80.2% and >37.12%, respectively. The differential KEGG pathway analysis showed that there were 17 differential functional genes, among which METABOLISM pathway-related genes accounted for 67.56%. In particular, Carbohydrate metabolism was significantly higher in August than in April (*p* = 0.028), while Amino acid metabolism pathway (*p* = 0.019) and Energy metabolism pathway (*p* = 0.003) were significantly lower. The differential COG function prediction analysis showed that there were nine differential functional genes, among which METABOLISM pathway-related genes accounted for 13.67%. In particular, Coenzyme transport and metabolism was significantly higher in April than in August (*p* = 0.03).

Among the 43 KEGG gene families and 24 COG functional genes identified in the April and December samples (rejuvenating and withered grass stages, respectively), METABOLISM pathway-related genes accounted for >80.6% and >37.16%, respectively. The differential KEGG pathway analysis showed that there were no significant differences in METABOLISM pathways. The differential COG function prediction analysis showed that there were three differential functional genes. In particular, the amino acid transport and metabolism pathway (7.63%) was significantly higher in April than in December (*p* = 0.018).

Among the 43 KEGG gene families and 24 COG functional genes identified in the August and December samples (green and withered grass stages, respectively), METABOLISM pathway-related genes accounted for >80.2% and >37.12%, respectively. The differential KEGG pathway analysis showed that there were 24 differential functional genes, among which METABOLISM pathway-related genes accounted for >57.4%. In particular, the amino acid metabolism pathway (6.89%) was significantly higher in December than in August (*p* = 0.019). The differential COG function prediction analysis showed that there were nine differential functional genes, among which METABOLISM pathway-related genes accounted for >15.90%. In particular, the amino acid transport and metabolism pathway (7.63%) was significantly higher in August than in December (*p* = 0.003).

### 3.4. Expression of Rumen Epithelial Nutrient Absorption- and Barrier-Related Genes in Different Months

The expression levels of nutrient absorption and barrier-related genes in the rumen epithelium in different months are shown in Figure 5. *CLAUDIN4* expression was higher than the expression of the other assessed genes in each month, and the trends in *CLAUDIN4* and *ZO1* were the same. *CLAUDIN4* expression was significantly higher in April than in August and December (3.04- and 30.9-fold, respectively). *ZO1* expression was 2.11- and 4.19-fold higher in April than in August and December, respectively. *SGLT1* expression was highest in August, which was 3.37- and 3.50-fold higher than that in April and December, respectively.

### 3.5. Interactions among Rumen Fermentation, the Microbiota, and Host Gene Expression in Tibetan Sheep

A correlation heatmap (correlation threshold > 0.5) of the top 20 species (with high abundances of their corresponding genera in the rumen of Tibetan sheep) and fermentation parameters (SCFAs and NH_3_-N levels and CL and ACX activities) and gene expression (2^−∆∆CT^) is shown in Figure 6. There were significant correlations between rumen fermentation parameters and genera (*p* < 0.05), with 33 positive and 20 negative correlations. Among these correlations, acetate was significantly positively correlated with *Fretibacterium*, *Ruminococcaceae_NK4A214_group*, and *Rikenellaceae_RC9_gut_group*. Propionate was significantly positively correlated with *Prevotellaceae_UCG-003*, and significantly negatively correlated with *Ruminococcus_1*. Butyrate was significantly positively correlated with *Prevotella_1* and *Succiniclasticum*, and significantly negatively correlated with *Rikenellaceae_RC9_gut_group* and *Ruminococcus_1*. NH_3_-N level was significantly positively correlated with *Ruminococcaceae_NK4A214_group* and *Succiniclasticum*, and significantly negatively correlated with *Ruminococcus_1* and *Rikenellaceae_RC9_gut_group*. ACX activity was significantly positively correlated with *Ruminococcus_1*, *Rikenellaceae_RC9_gut_group*, and *[Eubacterium]_coprostanoligenes_group*, and significantly negatively correlated with *Prevotellaceae_UCG-003*.

The expression levels of nutrient absorption- and barrier-related genes were significantly correlated with bacterial genera (*p* < 0.05), with 12 positive and 8 negative correlations. Among these correlations, *Ruminococcaceae_NK4A214_group* and *Succiniclasticum* were significantly positively correlated with *SGLT1* expression; *Rikenellaceae_RC9_gut_group* and *Ruminococcus_1* were significantly negatively correlated with *SGLT1* expression (*p* < 0.01); *Succiniclasticum* was significantly positively correlated with *CLAUDIN4* and *ZO1* expression; and *Fretibacterium* and *Rikenellaceae_RC9_gut_group* were significantly negatively correlated with *CLAUDIN4* and *ZO1* expression.

## 4. Discussion

As an important ruminant in the Qinghai-Tibet Plateau, Tibetan sheep graze naturally throughout the year, and mainly obtain nutrients by eating natural forages. The level of crude protein (CP) and ether extract (EE) in the forage grass in the green grass stage is significantly higher than that in the returning green stage and the withered grass stage, The level of acid detergent fiber (ADF) and neutral detergent fiber (NDF) and in the forage grass in the withered grass stage is significantly higher than that in the returning green stage and the green grass stage [2]. Regarding energy utilization, the A:P ratio is inversely proportional to the energy utilization efficiency of forage [22], indicating that the energy utilization efficiency of Tibetan sheep is lowest in December and highest in April. During the green grass stage, Tibetan sheep rapidly eat forages with sufficient nutrient levels within a small range [23], so metabolism decreases and more energy is stored in the body. In contrast, in the withered grass stage, Tibetan sheep need to travel across a larger area to search for forages, which increases metabolism and thereby decreases energy storage. In ruminants, >75% of SCFAs produced by rumen fermentation are absorbed via the rumen epithelium as the main host energy source [24]. We found that the total SCFAs, propionate and butyrate were higher in August and October, and the propionate and butyrate were significantly higher than in other months; NH_3_-N levels were significantly higher in June and August than in other months. Forages in the green grass stage have high protein and carbohydrate content [25], and Tibetan sheep can produce high NH_3_-N levels after eating them by increasing the level of the main rumen community in the rumen [2]. Butyrate is the main source of ruminant metabolic energy [26,27], and both propionate and butyrate can promote the development of rumen papilla [6]. August and October have higher rumen total SCFAs, propionate, butyrate and NH_3_-N concentrations, and rumen epithelial *SGLT1* expression is also significantly increased. This is due to the fact that the quantity and quality of the green grass stage forages have increased, and ACX activity has increased significantly, degrading more cellulose in subtilis into small molecules that can be used by the host. The NH_3_-N and total SCFAs levels produced by rumen microbial fermentation are increased, which produces more glucose. At this time, the high *SGLT1* expression in the rumen epithelium helps to transport D-glucose to the blood faster [28], reduce the glucose concentration in the rumen, and improve or prevent rumen acidosis [29]. It also increases the host energy utilization. Rumen microbes have adapted to highly nutritional forages, so the various cellulose-degrading bacteria in the rumen of Tibetan sheep reach a stable level by August, and CL activity remains at a stable level. Therefore, August is most conducive to the rumen development and body growth of Tibetan sheep.

The rumen microbes of Tibetan sheep and their hosts have evolved a relatively stable microbiota structure during long-term co-evolution in order to adapt to the harsh environment of the Qinghai-Tibet Plateau. Among the many factors affecting the rumen microbiota structure, diet is key [30]. In this study, the microbial abundances and diversity are significantly higher in August and October than in April and June. The analyses of KEGG gene families and COG functional genes of microorganisms found rumen microbes in different months adjust the energy utilization mechanism in different periods by influencing the carbohydrate metabolism pathway, amino acid metabolism pathway and energy metabolism pathway under the host’s metabolism pathway, which can affect host gene expression.

At the phylum level, both Bacteroidetes and Firmicutes were dominant (> 83.3%) in each month, and the abundance of Bacteroidetes was higher than that of Firmicutes. This is consistent with the results reported by Liu et al. [2]. However, Kim et al. [31] found that the mean abundances of Bacteroides and Firmicutes in the rumens of low-altitude livestock were only 31% and 56%, respectively. Cunha et al. [32] also found that the mean abundances of Bacteroides and Firmicutes in goats in semi-arid regions of Brazil were only about 37.9% and 56.3%, respectively. In the current study, there were no significant differences in the abundance of Bacteroidetes in different months, and it only increased in December. Members of this phylum can effectively decompose dietary protein and carbohydrates into SCFAs, providing energy for the host [33], and they also promote rumen growth and increase its volume [6]. The abundance of Firmicutes was significantly higher in June than in February and December. Firmicutes can help cells absorb glucose [6]. The increase in the abundance of Bacteroidetes in December indicates that it plays a more important role than Firmicutes in the high-altitude adaptation of the host during the period of nutrient deficiency. The abundances of Bacteroidetes and Firmicutes in the rumen of Tibetan sheep were maintained at stable levels during certain periods (from August to October) of the year, which is of great significance to the stability of the rumen internal environment. The increase in the abundance of Bacteroidetes in December may have promoted rumen development and forage decomposition, thereby providing a certain amount of energy for the host. The significant increase in the abundance of Firmicutes in June may have helped the host to obtain more energy.

At the genus level, *Prevotella_1* and *Rikenellaceae_RC9_gut_group* were dominant. *Prevotella_1* plays an important role in the degradation and utilization of plant non-cellulosic polysaccharides, including starch, xylan, and protein [22]. Reduced protein and starch levels decrease the abundance of *Prevotella_1* [34]. Studies have shown that high abundances of *Prevotella* and low abundances of *Methanobacter* in the rumen of Tibetan sheep promote forage fermentation to produce high SCFAs concentrations and reduce methane production to avoid energy loss [35]. *Rikenellaceae_RC9* is closely related to members of the *Alipites* family [36], and it may play a role in the degradation of plant-derived polysaccharides [37]. In this study, *Rikenellaceae_RC9_gut_group* had the highest abundance in December, which may have increased high-cellulose forage degradation, thereby providing energy for the host. *Rikenellaceae_RC9_gut_group* was a biomarker for December samples. Furthermore, random forest analysis showed that *Rikenellaceae_RC9_gut_group* played important roles in rumen microbial composition. The acetate concentration increased significantly in October and December, and the propionate and total SCFAs concentrations remained at high levels in October and December. The expression of *CLAUDIN4* and *ZO1* (which encode rumen epithelial barrier-related proteins that form a physical barrier around the cells to prevent the free passage of small molecules [38]) decreased significantly in December, that is, the permeability of the rumen epithelial barrier increased. In addition, many cellulolytic bacterial genera [39] were found in this study, such as *Ruminococcus_2*, *Fibrobacter*, *Butyrivibrio_2*, *Treponema_2*, and *Pseudobutyrivibrio*. Cellulolytic bacteria can degrade cellulose in the rumen and play key roles in SCFAs production [40]. Microorganisms have a rich library of protein-coding genes that can encode various enzymes related to metabolism [41]. Changes in the abundances of microbes lead to changes in enzymes, which alter the fermentation in the rumen [42]. ACX and CL activities dropped to its lowest level in August, ACX activity was significantly lower in August than in other months, while CL activity remained at a stable level (from April to December), which indicates that the digestion and utilization of cellulose in forage grass by ruminal microorganisms remained relatively stable. It also implies that after a long period of co-evolution, a relatively stable cellulolytic community exists in the rumen of Tibetan sheep. We speculate that the microorganisms encoding cellulase proteins are an important part of the “core microbiota” [43] in the rumen of Tibetan sheep, and cellulolytic bacteria play an important role in the plateau adaptation mechanism of Tibetan sheep.

Further study of the OTU that was found to be unique to the December samples (OTU270), as shown in the Venn diagram (Figure 1A), showed that the species belongs to the phylum *Synergistetes*, which mainly colonizes anaerobic environments [44] and can ferment amino acids and carry out glycolysis [45,46]. *Synergistetes* thoroughly and effectively decomposes forage via amino acid fermentation and glycolysis, thereby providing the host with as much energy as possible. *Synergistetes* has been reported to be a biomarker of periodontitis in the human oral cavity [47]. Therefore, we speculate that *Synergistetes* can increase the energy utilization of the host during the period of nutrient deficiency. In this situation, it is beneficial to the host. In December, when the quality of forage is further reduced, the abundance of *Synergistetes* in the rumen is significantly increased. Through the fermentation of amino acids and glycolysis, the forage can be decomposed more thoroughly and effectively, thereby providing the host with more energy. At this time, *CLAUDIN4* and *ZO1* expression also reached their lowest levels, the permeability of the rumen epithelium increased, more small molecules (mainly nutrient molecules) in the rumen easily pass through, and the host obtained more energy. However, when the abundance of *Synergistetes* increases to a certain level, the rumen epithelium of Tibetan sheep becomes inflamed, the barrier protein structure is gradually destroyed, and the rumen epithelial function decreases. To avoid long-term inflammation, after entering the rejuvenation stage (April), the host regulates the microbiota structure by actively regulating rumen epithelial genes, which significantly increases *CLAUDIN4* and *ZO1* expression, inhibits the passage of harmful substances, and returns the *CLAUDIN4* and *ZO1* expression to normal levels by August. This is a typical self-regulatory protection mechanism in Tibetan sheep in alpine regions with insufficient pasture. To prevent the decline in rumen epithelial function of Tibetan sheep observed in December, February, and April due to insufficient forage and the corresponding decrease in production performance, supplementary feeding should be carried out between December and April of the following year.

In addition, compared with the samples in December, February, and April, the number of differential species and biomarkers in the sample in June increased significantly (*p*
*<* 0.05), and the microbiota structure (the diversity and abundance of rumen microbes) changed significantly. Thus, compared to other months, the rumen microbial composition and structure of the June samples underwent substantial adjustments. Based on the KEGG and COG analyses, the METABOLIC pathway also underwent a certain degree of change from April to August. Therefore, we believe that June is the most critical transition stage for Tibetan sheep in the Qinghai-Tibet Plateau, which is of great significance to their growth and development.

## 5. Conclusions

The nutritional composition of forages in different periods of the Tibetan Plateau led to the changes in the abundance and diversity of rumen microorganisms of Tibetan sheep, which has resulted in the formation of different “core microbiota” in different periods, and then affected the fermentation pathway in the rumen, which was embodied in the changes in the content of fermentation products. After long-term co-evolution, Bacteroidetes, *Prevotella_1*, *Rikenellaceae_RC9_gut_group*, and cellulose-decomposing bacteria play important roles in the plateau adaptation of Tibetan sheep, which was mainly related to the mechanism of efficient energy utilization. The high expression of *SGLT1* gene in August is to transport more glucose to avoid rumen acidosis, and changes in the relative expression levels of *CLAUDIN4* and *ZO1* are associated with rumen microbial abundance to a certain extent. Therefore, the interactions between rumen fermentation products, the microbiota, and host gene expression of Tibetan sheep regulates the rumen fermentation pathway, mainly affecting the amino acid metabolism pathway and the energy metabolism pathway in the metabolism pathway, such that the microorganisms can completely decompose the cellulose in the forage and provide the host with as much energy as possible. At the same time, the balance of rumen “core microbiota” was regulated to promote the development of rumen and maintain the homeostasis of rumen environment. The interaction between fermentation products and microorganisms further affects the gene expression of the host and leads to changes in rumen epithelial permeability, so as to improve the energy utilization of the host and maintain the homeostasis of the rumen environment, which means that Tibetan sheep can better adapt to the harsh environment in different periods of the Qinghai-Tibet Plateau.

## Figures and Tables

**Figure 1 animals-11-03529-f001:**
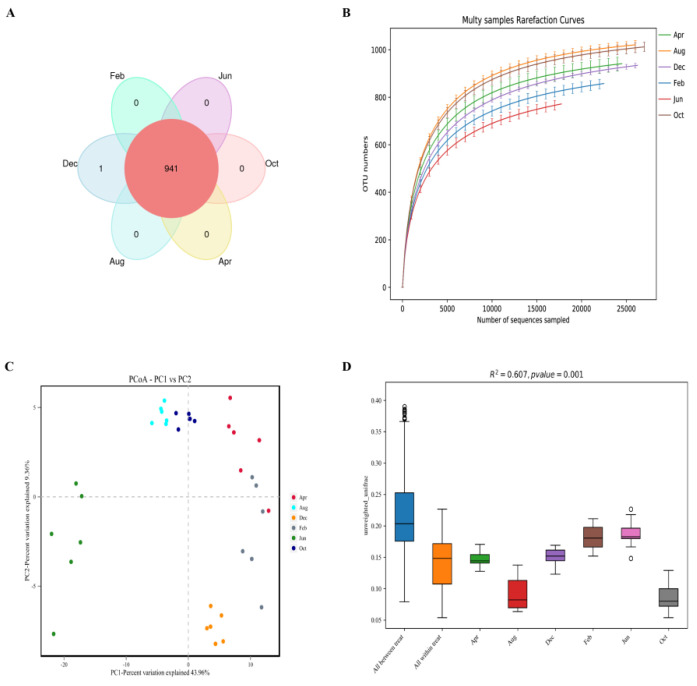
(**A**) Distribution map of microbial OTU; (**B**) dilution curve; (**C**) PCoA analysis; (**D**) Anosim analysis box plot.

**Figure 2 animals-11-03529-f002:**
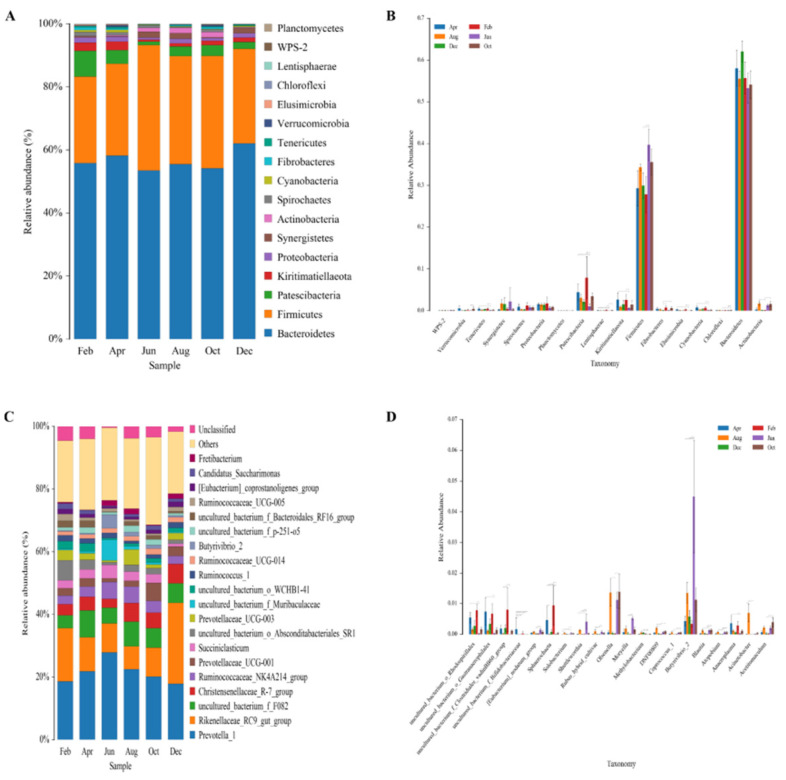
(**A**) Phylum species composition histogram; (**B**) phylum rank sum test analysis histogram; (**C**) genus species composition histogram; (**D**) genus rank sum test analysis histogram. Note: * *p* < 0.05.

**Figure 3 animals-11-03529-f003:**
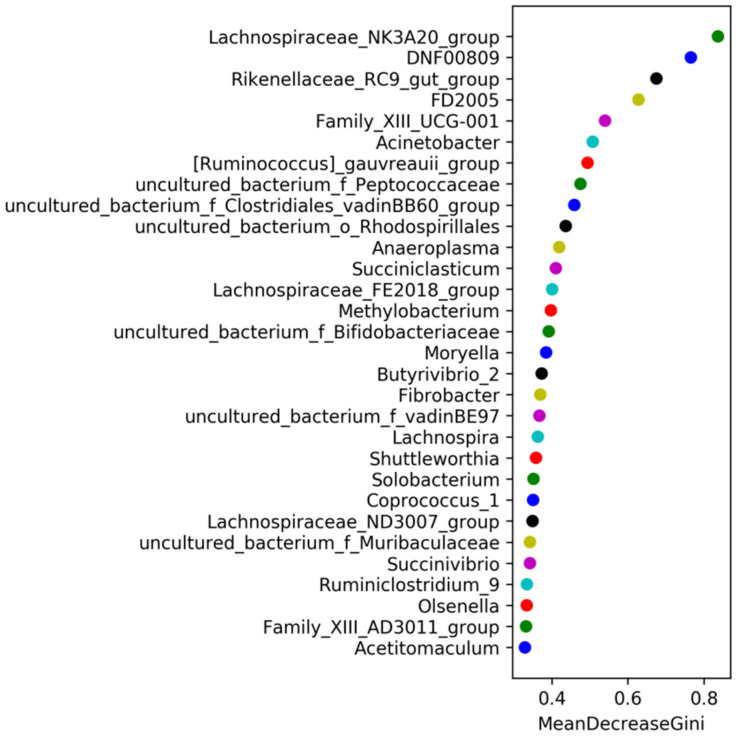
Random forest analysis of microorganisms at the genus level in different periods. Note: Mean Decrease Gini: Calculate the impact of each variable on the heterogeneity of the observations at each node of the classification tree, thereby comparing the importance of the variables. The larger the value, the greater the importance of the variable.

**Figure 4 animals-11-03529-f004:**
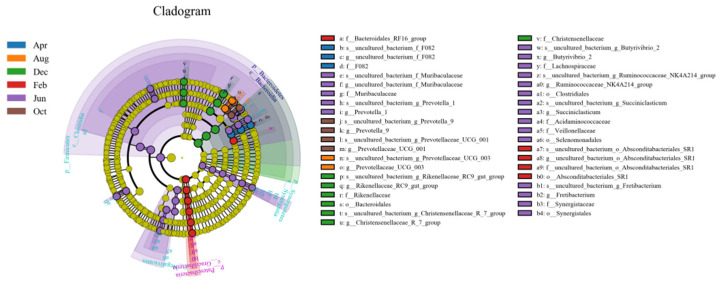
LEfSe analysis evolutionary branch diagram.

**Figure 5 animals-11-03529-f005:**
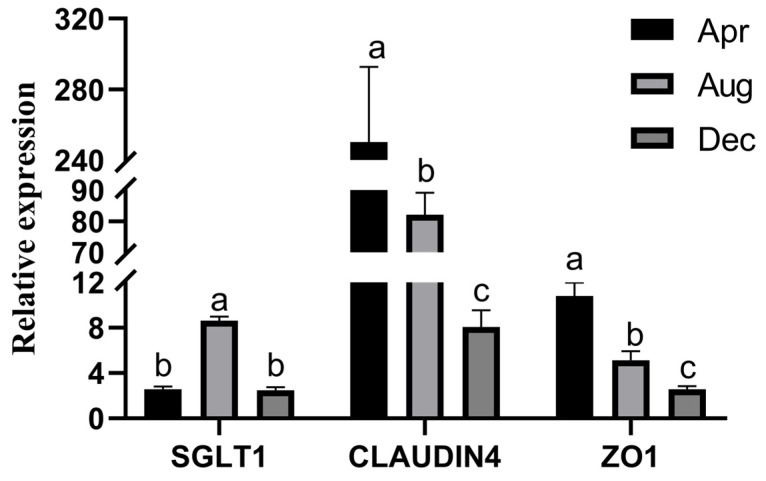
Expression of *SGLT1*, *CLAUDIN4,* and *ZO1* genes in rumen epithelial tissue. Note: Different lowercase letters represent significant differences in the relative expression of the same gene at different periods (*p* < 0.05). The same lowercase letter means that the difference is not significant (*p* > 0.05).

**Figure 6 animals-11-03529-f006:**
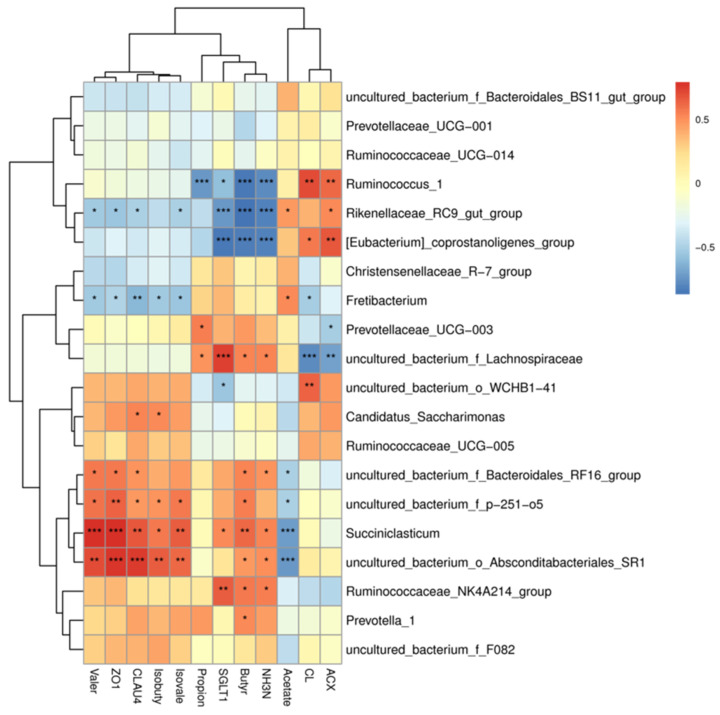
Rumen fermentation-microbe-host gene expression correlation heat map. Note: * *p* < 0.05, ** *p* < 0.01, *** *p* < 0.001.

**Table 1 animals-11-03529-t001:** Primer information.

Gene	Primer (5′-3′)	Length	Annealing Temperature	Login ID
*β-actin*	F:AGCCTTCCTTCCTGGGCATGGA	113 bp	60 °C	NM_001009784.3
	R:GGACAGCACCGTGTTGGCGTAGA			
*SGLT1*	F:GTGCAGTCAGCACAAAGTGG	198 bp	60 °C	NM_001009404.1
	R:CCCGGTTCCATAGGCAAACT			
*CLAUDIN4*	F:AAGGTGTACGACTCGCTGCT	237 bp	60 °C	NM_001185017.2
	R:GACGTTGTTAGCCGTCCAG			
*ZO1*	F:CGACCAGATCCTCAGGGTAA	161 bp	60 °C	XM_015101953.2
	R:AATCACCCACATCGGATTCT			

**Table 2 animals-11-03529-t002:** Results of determination of rumen fermentation parameters in different periods.

Fermentation Parameters	February	April	June	August	October	December
Acetate (mmol/L)	23.56 ± 0.33 ^d^	26.08 ± 0.81 ^c^	19.53 ± 1.94 ^e^	36.63 ± 0.33 ^b^	40.36 ± 0.88 ^a^	40.01 ± 2.01 ^a^
Propionate (mmol/L)	12.72 ± 0.61 ^c^	13.55 ± 1.02 ^bc^	10.33 ± 0.30 ^d^	14.37 ± 0.37 ^b^	15.96 ± 0.49 ^a^	13.86 ± 0.31 ^bc^
Butyrate (mmol/L)	6.85 ± 0.39 ^c^	7.48 ± 0.45 ^b^	5.71 ± 0.20 ^d^	8.11 ± 0.10 ^a^	8.14 ± 0.49 ^a^	6.05 ± 0.19 ^d^
Isobutyrate (mmol/L)	1.18 ± 0.11 ^a^	1.27 ± 0.09 ^a^	0.97 ± 0.02 ^b^	1.00 ± 0.01 ^b^	0.93 ± 0.05 ^bc^	0.82 ± 0.01 ^c^
Isovalerate (mmol/L)	2.53 ± 0.11 ^b^	2.77 ± 0.26 ^a^	1.85 ± 0.03 ^c^	1.87 ± 0.01 ^c^	1.69 ± 0.12 ^c^	1.44 ± 0.01 ^d^
Valerate (mmol/L)	5.05 ± 0.35 ^a^	4.61 ± 0.21 ^b^	4.26 ± 0.12 ^c^	4.13 ± 0.13 ^cd^	3.80 ± 0.06 ^d^	3.08 ± 0.15 ^e^
A:P	1.85 ± 0.06 ^c^	1.93 ± 0.08 ^c^	1.89 ± 0.13 ^c^	2.55 ± 0.04 ^b^	2.53 ± 0.02 ^b^	2.88 ± 0.08 ^a^
Total SCFAs (mmol/L)	51.88 ± 1.91 ^d^	55.77 ± 2.43 ^c^	42.65 ± 2.62 ^e^	66.11 ± 0.94 ^b^	70.91 ± 1.64 ^a^	65.27 ± 2.37 ^b^
NH_3_-N (mg/dL)	2.38 ± 0.03 ^e^	5.82 ± 0.08 ^c^	9.54 ± 0.26 ^a^	8.10 ± 0.02 ^b^	5.98 ± 0.03 ^c^	3.37 ± 0.03 ^d^
CL activities (μg/min/mL)	122.78 ± 17.25 ^a^	110.82 ± 15.58 ^ab^	110.13 ± 6.50 ^ab^	96.23 ± 2.01 ^b^	99.21 ± 4.85 ^b^	109.52 ± 5.58 ^ab^
ACX activities (nmol/min/mL)	144.61 ± 27.04 ^c^	293.32 ± 20.78 ^a^	257.4 ± 23.88 ^a^	107.64 ± 18.64 ^d^	162.19 ± 14.37 ^bc^	194.42 ± 13.55 ^b^

Note: In the same row, with different small letter superscripts mean significant difference (*p* < 0.05), the same lowercase letters or no letters indicate that the difference is not significant (*p >* 0.05).

**Table 3 animals-11-03529-t003:** Alpha index diversity.

Alpha Index	Shannon	Simpson	ACE	Chao1	OTU	Coverage (%)
February	5.24 ± 0.12 ^c^	0.020 ± 0.004	940.54 ± 13.38 ^c^	953.76 ± 14.19 ^d^	882 ^c^	99.60
April	5.58 ± 0.05 ^ab^	0.010 ± 0.001	1000.69 ± 24.72 ^b^	1015.56 ± 24.39 ^c^	952 ^b^	99.60
June	5.24 ± 0.12 ^c^	0.019 ± 0.007	891.69 ± 17.80 ^d^	909.56 ± 18.81 ^d^	819 ^d^	99.60
August	5.67 ± 0.08 ^a^	0.011 ± 0.001	1067.62 ± 16.55 ^a^	1076.28 ± 15.89 ^ab^	1027 ^a^	99.70
October	5.67 ± 0.07 ^a^	0.011 ± 0.002	1073.34 ± 14.22 ^a^	1092.61 ± 13.21 ^a^	1030 ^a^	99.70
December	5.35 ± 0.03 ^bc^	0.018 ± 0.002	1010.09 ± 11.01 ^b^	1031.25 ± 12.09 ^bc^	948 ^b^	99.60

Note: In the same column, with different small letter superscripts mean significant difference (*p* < 0.05), the same lowercase letters or no letters indicate that the difference is not significant (*p >* 0.05).

## Data Availability

The microbial sequences in our manuscript have been deposited in the Sequence Read Archive (SRA) of the NCBI, Accession No. SRR15033262-SRR15033297.

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
