# Peer review of "Rumen Fermentation—Microbiota—Host Gene Expression Interactions to Reveal the Adaptability of Tibetan Sheep in Different Periods"

_animals, 2021, doi:10.3390/ani11123529_

Round 1

Reviewer 1 Report

The manuscript entitled ‘Interactions Research Based on Among Rumen Fermentation Characteristics, the Microbiota, and Host Gene Expression to Reveal the Plateau Adaptability of Grazing Tibetan Sheep in Different Periods’ justifies publication in Animals, because topic is within the scope of the journal. The idea of the study is very inventive. The information presented in the manuscript is interesting and valuable. The amount of data presented is adequate to understand and interpret its scientific content (3 tables, 6 figures).

My comments and suggestions are presented below.

Title: I suggest to shorten the title. In this form is too long. Please improve it.

 The language of the manuscript needs some improvements.

L14: should be ‘which relies’;

L28: please add dot after ‘energy utilization’;

L61: maybe better will be using ‘protozoa and fungi’ term instead of ‘macroorganisms’;

L72: different size of font (‘Claudin-4’), please correct it;

L84: please delete ‘t’ from the sentence;

L95: please delete ‘were’ from the text;

L103-105: I do not know if I correctly understand you. Were the animals slaughtered in the next year of the experiment?

L137: should be ‘20µl’;

L147: different size of font (‘ribosome database’), please correct it;

Results: The results should be presented as XXXX, XXX, XX.X, X.XX and 0.XXX (for example 1350; 135; 13.5; 1.35; 0.135). In the present form there are too many numbers after comma. It is applied for all Tables throughout the manuscript.

L261,263: different size of font (‘Moryella’, ‘Shuttleworthia’, ‘Feb’, Apr’), please correct it;

L319: please delete word ‘assessed’ after ‘in each month’;

L358-363: different size of font, please improve it.

Do you have any results of the chemical composition of the samples of forage used during different months of the experiment? It would be necessary and helpful to discuss the obtained results. If yes, please add some more information about it to the manuscript.

L367: should be ‘is stored in the body’;

L373: should be ‘Aug’;

L378-379: maybe it would be better to revise the sentence in this manner ‘Aug and Oct have higher rumen total SCFAs, propionate, butyrate and NH3-N concentrations (…)’;

L381: different size of font (‘green grass stage’), please improve it;

L384: please exchange comma with a dot after ‘more glucose’ term;

L409-410: the sentence should be write down in this way ‘Members of this phylum can effectively decompose dietary protein and carbohydrates into SCFAs (…)’;

L462: ‘the abundance’;

L463: ‘Synergistetes’ should be written in italics when compare to the earlier usage of this word;

L480-483: please review wording;

L504: please add dot after ‘as possible’;

L505: should be ‘the balance of rumen’.

Reviewer 2 Report

1) Maybe the corresponding author adjust the role of some authors, “#” and “*” should be clearly noted.

2) Some small mistake should be correct before submit the revised manuscript. For example, “t” in line 84 page 2 is not necessary.

3) Abbreviation should be uniformed in the manuscript. For examples, “SCFA” in line 111 and 174.

4) Please give me your reason why the MN NucleoSpin 96 Soil kit (Macherey-Nagel, Germany) was chosen for microbial DNA extraction? That is, please clarify its advantage for DNA extraction.

5) In tables and figures, maybe it is better to note that what “a” “b” “c” and “d” represent for?

6) Font size was not uniformed in the manuscript. For examples, “Moryella and Shuttleworthia ” and “Feb and Apr”in line 261 and 264.

7) Many mistakes appear in the references, please amend them carefully before submitting the revised manuscript. For example, “Frontiers in microbiology”in line 532 and “Front Microbiol.” in line 564 are the same journal? Furthermore, the references 8 and 9 are the same. Please uniform the style of all the references.

Reviewer 3 Report

This is an interesting topic to study the relationship between seasonal plant phenological change on plateau rumen animal's microbiome. But the study material, Tibatan sheep, study design, and the demonstration of results are highly similar to the paper published in "frontier in Microbiology" (doi: 10.3389/fmicb.2020.587558). It's a pity. If you can focus on seasonal phenological change on physiological and rumen microbiome change of Tibatan sheep, you will get more novel finding.

Some minor comments on the content.

Line 14, it's microbiome was known as "second genome", not microbes.

Line 164, ANCOVA is "Aanlysis of Covariance.

Figure 2D, I think you put a wrong figure. The one you put is almost the same with 2C.

Round 2

Reviewer 2 Report

The authors revised the manuscript according to reviewers’ comments. Furthermore, the authors changed the title “Rumen fermentation - microbiota - host gene expression inter-actions to reveal the adaptability of Tibetan sheep in different periods”. In my opinion, the revised manuscript could be accepted for publication before carefully checking the whole paper according to “Instructions for Authors”.

Reviewer 3 Report

The authors have revised the manuscript as comments. 

I have no further comment.